# Allosteric Binding Sites On Nuclear Receptors: Focus On Drug Efficacy and Selectivity

**DOI:** 10.3390/ijms21020534

**Published:** 2020-01-14

**Authors:** André Fischer, Martin Smieško

**Affiliations:** Computational Pharmacy, Department of Pharmaceutical Sciences, University of Basel, Klingelbergstrasse 50, CH-4056 Basel, Switzerland; and.fischer@unibas.ch

**Keywords:** nuclear receptor, allosteric site, molecular dynamics, docking, computational chemistry

## Abstract

Nuclear receptors (NRs) are highly relevant drug targets in major indications such as oncologic, metabolic, reproductive, and immunologic diseases. However, currently, marketed drugs designed towards the orthosteric binding site of NRs often suffer from resistance mechanisms and poor selectivity. The identification of two superficial allosteric sites, activation function-2 (AF-2) and binding function-3 (BF-3), as novel drug targets sparked the development of inhibitors, while selectivity concerns due to a high conservation degree remained. To determine important pharmacophores and hydration sites among AF-2 and BF-3 of eight hormonal NRs, we systematically analyzed over 10 μs of molecular dynamics simulations including simulations in explicit water and solvent mixtures. In addition, a library of over 300 allosteric inhibitors was evaluated by molecular docking. Based on our results, we suggest the BF-3 site to offer a higher potential for drug selectivity as opposed to the AF-2 site that is more conserved among the selected receptors. Detected similarities among the AF-2 sites of various NRs urge for a broader selectivity assessment in future studies. In combination with the Supplementary Material, this work provides a foundation to improve both selectivity and potency of allosteric inhibitors in a rational manner and increase the therapeutic applicability of this promising compound class.

## 1. Introduction

Nuclear receptors (NRs) are ligand-inducible transcription factors that are attractive drug targets due to their involvement in a multitude of physiological and pathological processes. Currently marketed drugs designed to interact with the buried ligand binding pocket (LBP) of the respective receptor are used in major indications such as oncologic, metabolic, reproductive, and immunologic diseases [1,2]. However, the success of these therapeutics is often limited by poor selectivity and resistance mechanisms that, in the worst case, reverse the antagonistic effect of a drug and promote disease [3,4,5]. Additionally, undesirable effects are promoted by the fact that both inhibitors and natural substrates share the same binding pocket. In recent years, two allosteric sites on the surface of several NRs, called activation function-2 (AF-2) and binding function-3 (BF-3), have been identified and considered as alternative sites for drug binding (Figure 1A). The AF-2 site corresponds to a protein–protein interaction surface for the binding of coactivator proteins essential for downstream signaling, which renders it an attractive target for potential inhibitors. While the BF-3 site has been initially shown to allosterically regulate binding of coactivators to the AF-2 site [2,6,7,8], it has been suggested as an interaction surface for the engagement with chaperones that associate NRs [2,9,10]. In recent years, several hundreds of compounds have been identified to modulate NR activity through either of these allosteric sites at various receptors [5,6,7,8,11,12,13,14,15,16,17,18,19,20,21,22,23,24,25,26,27]. Selectivity testing in the mentioned projects was, if conducted, in most cases, limited to a single other NR [13,16,19]. Since especially steroidal NRs such as androgen receptor (AR), estrogen receptors (ER), glucocorticoid receptor (GR), progesterone receptor (PR), and minearlocorticoid receptor (MR) share a common domain architecture as well as a similar fold regarding their ligand binding domain (LBD), the selectivity concern for allosteric NR inhibitors remains (Figure 1B). For example, it has been reported that the AF-2 and BF-3 sites of AR and GR have a sequence identity of approximately 50% [9,28]. Further on, drug-like mimetics of coactivator peptides at the AF-2 site have the potential to disrupt protein–protein interactions for multiple NRs and, ultimately, promote off-target toxicity [16,29].

While several structures with cocrystallized ligands have been determined for the AR, targeting superficial binding sites such as AF-2 and BF-3 of other receptors remains a challenge due to their comparably large size, shallowness, and high flexibility [30,31]. Knowledge regarding binding hotspots and distinct pharmacophores among structurally similar NRs is crucial for the design of effective and selective inhibitory compounds [31,32,33,34]. In this regard, cosolvent molecular dynamics (MD) simulations are a suitable computational tool to determine hotspots and assess their druggability, as well as to obtain detailed information on potentially useful pharmacophores to improve drug potency and selectivity for the site of interest. In this simulation protocol, which was inspired by crystallographic observations of small fragments binding to protein surfaces, organic solvent molecules mimicking drug fragments are added to the aqueous phase to monitor and quantify their interaction with a protein. Compared to similar methods for binding site detection, this protocol depends less on the input structure, allows for conformational changes of the protein, and is, generally, more reliable due to the intrinsic treatment of protein flexibility and explicit solvation [31,32,35,36]. Even though cosolvent simulations have been previously applied to study the allosteric sites of AR, ERα, and ERβ, the main objective of these studies was to proove the applicability of the simulation protocol. In two studies that considered the AR, researchers were able to detect both allosteric sites in the top hotspots and, based on an assessment regarding the maximum achievable binding affinity, the AF-2 site was deemed more druggable [31,37,38]. Another known drawback of solvent-exposed binding sites is the influence of water molecules on the recognition, efficacy and selectivity of ligands due to their potential displacement or mediation of ligand–protein interactions. Whether a water molecule can be favourably displaced depends on its environment in the respective binding site [39,40,41,42]. The desolvation free energy of a water molecule can be estimated based on MD simulations followed by the quantitative assessment of the trajectory snapshots and can, ultimately, guide the design of novel compounds or improve scoring in virtual screening projects [41,42,43].

Here, we applied cosolvent MD simulations, hydration site prediction, and molecular docking to a set of eight NRs including AR, ERα, ERβ, GR, MR, PR, and the thyroid receptors α and β (TRα and TRβ). We assessed a large share of compounds reported to modulate NR signaling through either AF-2 or BF-3 of the respective receptor.

In contrast to previous works, our objective was the systematic determination of the main pharmacophores and positions of structural waters in order to compare them within our selection of human hormonal NRs to ultimately navigate the design of potent and selective inhibitors for each particular receptor.

## 2. Results and Discussion

### 2.1. Sequence Similarity Among Hormonal NRs

The structures of LBD of AR, ERα, ERβ, GR, MR, PR, TRα, and TRβ feature a similar conserved fold (Figure 1B). We conducted a sequence-based analysis of residues in the 5 Å range around a cocrystallized ligand and compared those residues with respect to their similarity among all receptors (Figure 2). While the analysis revealed similarities between the isoforms of the ERs and TRs that were expected, receptors with high identity in both sites included AR, GR, MR, and PR. We observed values up to 75% between MR and GR, as well as PR and AR, in the AF-2, which raises serious concerns regarding off-target binding for ligands targeting either of these receptors. In slight contrast to the previously reported conservation degree in the literature [2,9], our comparably high percentages could be explained by the different definitions of conservation and binding site residues. Compared to the other receptors, the TRs offer a good potential for selective binding to either site, but particularly, for BF-3. Overall, the results suggest that the BF-3 site offers a higher potential for the design of selective inhibitors due to the generally lower values in similarity among the receptors compared to the AF-2. The conservation of residues from a three-dimensional perspective can be assessed in Appendix A.

### 2.2. Distinct Pharmacophores of the Allosteric Sites

To this date, the AR is the most intensively studied NR regarding the development of allosteric inhibitors, due to its involvement in the genesis and progression of prostate cancer, which is one of the leading causes of cancer-related death in men. [2,13]. Even though constitutively active splice variants of the AR lacking the LBD regularly arise in late stages of the disease, the AR LBD remains a drug target of high interest, particularly in early stages of pharmacological treatment [2]. Likewise, efforts were put into the design of inhibitors against ERα since the majority of breast cancer cases depend on this receptor [24]. Unfortunately, currently available therapeutics often suffer from resistance mechanisms, in some cases, only caused by as little as a single amino acid mutation in the LBP [4], which likely contributed to the number of works that applied cosolvent simulations to the allosteric sites of the AR and the ERs. In these studies, the AF-2 site was detected in both AR and ERs, while densities at the BF-3 site were only reviewed for the AR [31,37,38].

The aforementioned work inspired us to systematically apply this simulation protocol to eight NRs which are known to suffer from poor drug selectivity [2,24,27]. Based on the evaluation of our simulations, we were able to identify probe molecules binding to the AF-2 and BF-3 sites of all receptors, with the exception of the BF-3 site in ERα (Figure 1C and Figure 3A). In accordance with the sequence analysis, the similarity among the AF-2 sites regarding the probe densities of all receptors along with the diversity of the individual BF-3 sites was one of the most apparent outcomes of our simulations. The results do not only reflect the preference of multiple NRs for similar coactivator sequences, a known concern for receptor selectivity [45], but also support the fact that a higher degree of selectivity could be achieved when targeting the BF-3 site over both the orthosteric pocket and the AF-2 site due to its uniqueness among the receptors. Clearly, the selectivity concerns regarding inhibitors interacting with the AF-2 site were justified because simulations of GR, MR, PR, and the the TRs, in particular, presented a highly similar pattern of probe densities. However, despite the comparably high sequence similarity of GR, MR, and PR regarding the AF-2 site, the GR resulted in a notably higher density of acetonitrile, which points towards a higher degree of amiphaticity that is favored there. Interestingly, the ERs not only displayed distinct differences to the other receptors, but also between themselves based on two isolated densities of isopropanol and pyrimidine that were interchanged between the two isoforms. Even though it is possible that compounds assume reversed binding modes in either receptor, such distinct differences offer the potential to improve isoform selectivity, particularly if structure-based design is employed. Less evidently, the density map of the ERα revealed a smaller third hotspot in the vicinity of V368 unique to this receptor, suggesting this to be the reason for the selectivity differences regarding AR and ERα observed among particularly decorated inhibitors with a common pyrimidine core [10]. In consensus with that, the AR displayed an isolated density of pyrimidine towards H4 and a generally more pronounced aromatic density. The higher aromaticity of the AR AF-2 site compared to the ERs likely reflects its preference for phenylalanine or tryptophan residues as opposed to leucines in coactivator fragments [15]. Notably, simultaneous binding to multiple NRs can be desired in certain therapeutic scenarios as for the inhibition of the AR in breast cancer treatment, since it might adopt roles of ERα when the function of this receptor is absent due to pharmacological inhibition [46]. Therefore, compounds binding to both AR and ERα, such as several ones identified by Gunther et al., might be beneficial, depending on the therapeutic indication [2,14]. While the literature suggests differences in the hydrophobic relief of the AF-2 to be responsible for TR isoform selectivity [26], our results only presented minor differences in any of the probe densities between TRα and TRβ, apart from a slightly oblonged density for pyrimidine in the TRβ.

As mentioned before, the BF-3 site of the studied NRs displayed a higher degree of heterogeneity regarding the cosolvent densities (Figure 3A). While AR, GR, and to some degree MR and PR showed a somehwat comparable pattern of probe densities, both ERs and TRs presented a high degree of diversity among them despite being most closely related based on their sequence. Most evidently, the BF-3 site of ERα was barely mapped by the probe molecules at the selected isovalue (twelve times the density in bulk solvent) and only presented a slight density of pyrimidine, indicating a region for an aromatic moiety. The same region in ERβ was mapped by isopropanol, suggesting the placement of an amphipathic, as opposed to an aromatic, functional group in this isoform, to achieve selective ligand–protein interactions. The lack of probe density at the ERα BF-3 site points towards poor druggability of the site [47], which is indirectly supported by experimental results since inhibitors directed against the AR did not inhibit ERα [13]. The density maps of the BF-3 site of both TRs substantially differ from the ones of other receptors, which reduces the odds for the cross-binding of compounds harboring the proposed density-based pharmacophores. Even though the TRβ shared densities for isopropanol and acetonitrile with the TRα, the latter displays additional densities in distal regions of the site and a pronounced density of acetonitrile in the center. Therefore, an amphipathic group, potentially containing a nitrogen atom, would offer the potential to increase compound selectivity between the two TRs. Moreover, the density maps revealed a high identity between AR and GR, particularly regarding the regions mapped by isopropanol. In contrast, the MR and PR presented distinct differences despite the comparable degree of conservation among all four receptors.

A comparison of the AF-2 and BF-3 displayed density patterns that are shared among the two sites depending on the receptor. For example, the densities for the ERα AF-2 and PR BF-3 site showed a similar arrangement consisting of an amphipathic group coupled to an aromatic moiety. Furthermore, the BF-3 sites of both AR and GR showed a distinct resemblance to the AF-2 sites of most other receptors. The consideration of compounds designed for the AF-2 site to simultaneously interact with the BF-3 site and vice versa is not only supported by our results, but is also based on crystallographic data. Most interestingly, a crystal structure of the AR (PDB ID: 2YLP) [12] revealed two ligands concurrently bound to both allosteric sites. Therefore, a complete selectivity assessment should consider binding to the other allosteric site as well. In addition to the allosteric sites, we measured comparably high probe densities in several orthosteric sites, regions that were suggested to be involved in the access to the buried binding pocket of NRs, and other zones of the receptors [48]. For detailed review and to assist the development of novel compounds [49], we supply the complete density maps for every receptor in our Appendix A. The root mean square deviation (RMSD) of the cosolvent simulations was assessed (Appendix A) and presented deviations ranging from 0.81 to 2.57 Å, confirming good conformational stability of the protein backbone throughout the simulations.

### 2.3. Conformational Changes of the Allosteric Sites

Even though it was suggested that the association of allosteric inhibitors is dependent on the presence of an agonist in the orthosteric site [23], we did not observe significant differences regarding the cosolvent densities between our apo and holo simulations (Appendix A). Potentially, a protocol with prolonged individual simulations or the application of biasing potentials might induce more pronounced changes, since conformational adaptations affecting the surface of the receptor have to occur over a long distance and, naturally, require substantial simulation efforts [36]. For example, association of inhibitors to the LBP has been shown to structurally modulate the AF-2 and its capability to interact with coactivator proteins, mainly by conformational change of helix-12 (H12) [10]. Combination therapy with multiple drugs is regularly applied in cancer pharmacotherapy [50,51] and, therefore, potential synergistic effects of allosteric and orthosteric inhibitors will have to be considered in future studies. Likewise, the simultaneous treatment with AF-2 and BF-3 inhibitors might produce mixed results, since binding of inhibitors to the BF-3 site is known to reduce the affinity of coactivator peptides in an allosteric mechanism and might affect a potential drug–drug synergy. In general, the AF-2 site of NRs is known to be capable of significant conformational changes as the example of the AR nicely underlines, since this receptor accepts a diverse set of coactivator fragments and has to structurally adapt in order to do so [15,20]. Our examination of available AR crystal structures (Appendix A) revealed certain residues in both AF-2 and BF-3 capable of structural adaptation to ligand molecules. In particular, the residues K720, R726, and M734 in the AF-2 displayed various rotamers depending on the interaction partner. In the BF-3 site, Q670, F826, N727, E829, K836, and R840 appeared flexible, suggesting this site exhibits a higher degree of flexibility.

To quantify the conformational change induced by different probe molecules in either allosteric site, we compared representative structures of our simulations in pure water to ones obtained from cosolvent simulations (Appendix A). In accordance with the above-mentioned flexibility in crystal structures, this analysis uncovered a higher degree of structural adaptation of the BF-3, as opposed to the AF-2. In this context, it is worth noting that several residues of the BF-3 site are located in the vicinity of the N-terminus, which would likely be more rigid due to its direct linkage to the DNA-binding domain that was not considered here. Probe molecules in cosolvent simulations have been shown to induce cryptic binding pockets, which can significantly contribute to drug selectivity and, therefore, knowledge on such pharmacophores can be instrumental for rational drug design. It is known that individual cosolvent molecules can cause distinct conformational adaptations of the respective protein [36,52]. Indeed, our RMSD-based analysis presented the highest degree of structural change with isopropanol as a probe molecule, suggesting the associated pharmacophores as promising to exploit the intrinsic flexibility of both allosteric pockets. Interestingly, we observed a different extent of adaptation from receptor to receptor, with the ERα showing the most distinct changes. In the literature, residues with a high degree of flexibility have been reported for the AR and include K720, M734, N727, F826, E829, and F837 [11,12]. Although our analysis detected several of these residues to be involved in a comparably large structural adaptation, we observed conformational changes of additional residues that have not been reported before. Most notably, the residue L685 in the GR (corresponding to F826 in the AR) located in the BF-3 site displayed a particularly high RMSD. A large share of allosteric NR inhibitors have been developed based on the interplay of computational screening and experimental characterization. Importantly, our results could support future virtual screening studies by recommending flexible residues in the binding site for molecular docking calculations. One limitation we experienced during this analysis was the truncation of termini in several crystal structures, preventing a quantification of the RMSD for these regions and adjacent structural elements. The backbone RMSD analysis (Appendix A) of all triplicates simulations in pure water revealed excellent (AR, ERα, GR, PR, TRβ) to sufficient (ERβ, MR, TRα) convergence.

### 2.4. Displacement of Water Molecules from the Allosteric Sites

In addition to their often limited selectivity, the major drawback for the clinical application of allosteric inhibitors is their comparably low potency, which is a typical characteristic of superficial protein–protein interaction inhibitors [18,53]. Regarding the ERα, it was proposed that the modest efficacy of small molecules designed to bind to the AF-2 surface is also associated with the lower amount of water molecules that are displaced by the inhibitor in contrast to the physiologic interaction partner [22]. As mentioned in the introduction, the importance of considering water molecules is well known in structure-based drug design because, in almost any case, waters are displaced during ligand association. Displacing a tightly bound water molecule generally results in a favorable gain of entropy and is a commonly used strategy exploited by medicinal chemists to improve compound efficacy and selectivity [39,40,54]. However, if such a displacement is favorable depends on enthalpic and entropic contributions that are determined by the environment of the water molecule. Ultimately, the desolvation free energy of a water molecule can be estimated based on these contributions and provide a valuable input for ligand design [42]. Particularly for solvent-exposed binding sites, as they often are when protein–protein interactions are considered, contributions of water molecules can gain even higher importance [22].

We evaluated the hydration sites for the selected set of NRs by two different approaches. The WATsite program allows a restrained MD simulation in explicit water to be set up, which is post-processed using machine learning techniques to estimate the thermodynamic contributions of a water molecule to be displaced. Following this analysis, we retrieved available crystal structures for every receptor and determined conserved hydration sites among them (Appendix A) to ultimately combine both predictions and find a consensus. Although a large share of the water positions we determined were unique to each NR, there were sites shared by multiple receptors (Figure 1C, Figure 3A and Appendix A). For example, we identified a conserved water molecule at the AF-2 site of both isoforms of the ER that could be favorably displaced based on its desolvation enthalpy (Appendix A). Another interesting example was a water molecule with a negative enthalpic contribution and conservation within crystal structures close to H12 that was predicted to occur in only one isoform of the TR (Figure 1C). Therefore, displacing this water molecule with a TRβ AF-2 inhibitor might increase the selectivity towards TRα. In a similar fashion, displacing a conserved water molecule that occurs in AR, ERβ, GR, and MR from a favorable environment might decrease binding of a compound to any of these receptors (Appendix A). Remarkably, no water molecules with a negative enthalpic contribution were identified for the AF-2 site of the PR. At the BF-3, we observed a higher diversity of hydration sites compared to the AF-2 site, following the previous trends regarding our conservation analysis and the cosolvent densities. However, we noticed a reoccurring network of water molecules in the vicinity of the H9 N-terminus that, depending on the compound, might form a favorable first-shell hydration layer based on enthalpic contributions (Figure 3C) [43]. The simulations showed small backbone fluctuation values, which was to be expected due to the applied restraints on the protein atoms (Appendix A). Analogous to our cosolvent simulations, structure files resulting from both procedures used for hydration prediction, together with a complete list of enthalpic and entropic contributions for each water molecule (Appendix A), are provided in the Supporting Material. These contributions can be used to estimate the gain in free energy of a particular ligand molecule by considering the water molecules it displaces in the bound state.

### 2.5. Selectivity of Allosteric Inhibitors Explored by Molecular Docking

The design of numerous allosteric inhibitors considered in this study was itself assisted by computational chemistry methods such as virtual screening [12,13,17,24]. Molecular docking is an accepted technique with high throughput to explore off-target binding of potential drug compounds, preferably in an early stage of their development [55,56]. Here, we retrieved more than 300 confirmed allosteric inhibitors from the literature and cross-docked them to investigate their potential to interact with the AF-2 and BF-3 of other NRs in our selection. Independent of the site towards which the inhibitors were designed, they were docked to both allosteric sites, because most studies experimentally excluded an LBP-based mechanism, but did not distinguish between them [4,24,25]. As a common practice, we assessed the accuracy in pose prediction by redocking cocrystallized ligands to the respective site (Appendix A). The determined heavy-atom RMSD values between the poses reached from 1.25 to 7.46 Å for the standard precision (SP) docking protocol and from 0.84 to 8.22 Å for the extra precision (XP) protocol in Glide (Appendix A) [57,58]. A visual inspection of the obtained poses revealed a reversed orientation for multiple ligands at the BF-3 site, which might be explained by the symmetry of several compounds regarding their aromatic moieties, as well as the increased conformational freedom at such solvent-exposed binding sites. In addition, the probe densities from our cosolvent simulations (Figure 1C and Figure 3A) justify a reversed orientation of the main pharmacophores in certain cases. Another complication, potentially causing inaccuracies in pose prediction, is the presence of crystal mates in close vicinity to the cocrystallized ligand [59], which we determined in various crystal structures as demonstrated in Appendix A. Despite a more sophisticated scoring function and the considerably higher computational cost of the XP protocol, it did not offer any obvious improvement regarding pose prediction in the selected cases. Additionally, the Glide SP protocol was successfully used for the determination of inhibitors towards the AF-2 and BF-3 sites of the AR [11,17,21,22] and we, therefore, selected it to evaluate our library of ligands. To further validate the performance of the chosen docking protocol, we generated a decoy ligand set and determined the area under the curve of the Receiver Operator Characteristic (ROC AUC) for each compound group. Based on a poor score in this metric (Appendix A), compounds designed towards the AF-2 site of TRα and TRβ were excluded from further investigation. An analysis of the score distribution was conducted for the remaining compounds series with more than ten entries, meaning ligands designed towards the AF-2 sites of AR and ERα, as well as the BF-3 of the AR, were docked into their target site to compare their scores to the ones of all other compounds in our ligand set. Between the allosteric sites of the AR, a distinct difference regarding the score distribution could be observed (Figure 3D,E). While the distribution of docking scores of the AF-2 compound series showed a high overlap, the curves displayed an astounding degree of separation for the BF-3 site. The largest share of compounds designed to interact with the BF-3 site of the AR showed an average improvement in binding free energy of approximately 1.0 kcal/mol, with a high number of the remaining compounds scoring below −3.0 kcal/mol. Again, these results suggest targeting the BF-3 site to offer a higher degree of selectivity compared to the AF-2 site, particularly since the distribution of the ERα AF-2 compound series presented a similar pattern as the AR AF-2 series (Appendix A). The score distribution obtained from the XP docking protocol (Appendix A) confirmed the results obtained from the SP protocol.

Even though the majority of studies neglected off-target binding of their compounds, one study extensively evaluated their inhibitors against other NRs overlapping with our selection [24]. Although their ERα ligand showed reasonable selectivity against AR and PR, two receptors presenting a high degree of similarity throughout our work, it was shown to interact with the GR to a fair extent. Inspired by these results, we reviewed the docking poses of the compound and discovered a halogen bonding interaction [40] between a lysine residue, which is part of a so-called charge clamp, and the chlorine atom of the inhibitor shared by both GR and ERα (Appendix A). The charge clamps, flanking the hydrophobic subpockets of the AF-2 sites in various NRs, were often proposed as a selectivity factor [10]. The described interaction did not appear in either AR or PR and we, therefore, suggest this specific interaction as a determinant for the selectivity of this compound.

## 3. Materials and Methods

### 3.1. Sequence Alignment and Analysis

After a sequence alignment in the UGENE v1.32.0 [60] suite using the ClustalW algorithm [61] (Appendix A), we determined residues of either site based on a spherical zone around a cocrystallized ligand (PDB ID: 2YLP) that can bind both AF-2 and BF-3 in the AR. We then used an in-house python routine to determine the conservation of the selected residues as follows: identical residues were valued at 1.0, while the same residue group was valued at 0.5 (Appendix A). The values were summed up for 20 AF-2 residues and 23 BF-3 residues, respectively, to ultimately calculate a percentage value for the conservation.

### 3.2. Ligand Preparation

All ligands were retrieved from various publications that evaluated their compounds and provided evidence for binding to either of the allosteric sites (Appendix A). Compounds were included if a reasonable biological activity ($IC_{50}$ or $K_{i}$ below 100 μM) was measured. Three-dimensional conformers were generated in the LigPrep panel [62] within the Maestro Small-Molecule Drug Discovery Suite 2019-3 [63] using the OPLS3e force field. The protonation states of the ligands were predicted using Epik [64] at physiological pH (pH = 7.4). The highest scored ligand conformations were selected, potential tautomerization was accounted for, and in the case of unspecified chiral centers, both stereoisomers were considered.

### 3.3. Protein Preparation

The protein structures used in this study were retrieved from the Protein Data Bank (Appendix A) and prepared using the Protein Preparation Wizard [65] within Maestro. In the case of missing loops, they were added based on complete template structures for ERα, ERβ, MR, and TRβ. The amino acid sequence of the protein structures was compared to the sequence reported in the UniProt database [66] and, in the case of engineered amino acids, the sequence was manually corrected to represent the wild-type receptor sequence. For MD simulations, we aimed to select structures with physiological ligands bound to the LBP and a resolution below 2.5 Å. In the case of less than five missing amino acids at the C-terminus to complete the sequence, these residues were manually added to the structure. While the N-terminus was modeled with an acetamide cap, since it would be further linked to the DNA-binding domain, the C-terminus was modeled as free carboxylic acid group. Ions and organic solvents were removed, before hydrogen atoms were added to the structures, the protonation state predicted at pH 7.4, and the hydrogen bonding network was oriented. As a last step, the structures were refined by means of a restrained minimization using the OPLS3e force field with a RMSD convergence threshold of 0.30 Å.

### 3.4. MD Simulations and Evaluation

The simulations in pure water were conducted using the Desmond simulation engine (v.2019-3) [67]. Using the System Builder, the prepared protein structures were solvated with SPC water molecules in a cubic periodic boundary system with a buffer of 10 Å to the next protein atom. Ions were added to neutralize the systems, before they were relaxed for 100 ps using the MD-based Desmond Minimization protocol. The simulations were conducted using the OPLS_2005 force field in an NPT ensemble combined with the Martyna–Tobias–Klein barostat, with a relaxation time of 2.0 ps at 300 K and the Nose–Hoover thermostat, with a relaxation time of 1.0 ps. The u-series algorithm was used to treat long-range interactions with a cutoff of 9 Å for short-range interactions [68]. By default, the M-SHAKE algorithm was applied to constrain bonds to hydrogen atoms. We left the time step for the RESPA integrator at 2.0 fs and files with atomic coordinates were saved at an interval of 4.8 ps. After the default relaxation protocol (Appendix A), the simulations were carried out in triplicate for a duration of 40 ns per receptor at a temperature of 300 K and, to ensure a unique course of the individual trajectories, we generated random seeds for the initial velocities. The backbone RMSD of the pure water MD simulations were determined in the Simulation Interaction Diagram panel within Maestro.

For the cosolvent MD simulations, we used the Mixed Solvent MD workflow that comes with the Desmond simulation engine [67]. As probe molecules, isopropanol, acetonitrile, and pyrimidine at a concentration of 5% (by volume) were selected since these solvents are water-miscible, offer a low potential for aggregation, and, therefore, do not require the application of repulsive forces [34,69]. In addition to the recommended simulation protocol with apo structures, we ran simulations with the cocrystallized ligand remaining in the orthosteric binding pocket. For the ERβ and GR, the water buffer parameter was increased from the default value of 12.0 to 15.0, as described in the provided documentation. The default relaxation protocol for this workflow (Appendix A) was conducted, before the 5 ns production simulations were run at a temperature of 300 K in an NPT ensemble using the OPLS_2005 force field. For each probe molecule, ten simulations were conducted, resulting in a cumulative simulation time of 1.2 μs per receptor. The remaining specifications were left as the defaults. The backbone RMSD of the cosolvent MD simulations were determined based on the output frame of the protocol using an in-house python routine.

To quantify the conformational change of the AF-2 and BF-3 sites induced by the probe molecules, we determined the heavy-atom RMSD between representative structures of the cosolvent simulations and the pure water simulations of the individual residues located in the allosteric sites. First, we determined the representative structure of the cosolvent simulations by inputting the last frame of each simulation for each probe into the MaxCluster algorithm and selecting the structure with the highest rank according to the 3D-jury score [70]. Similarly, we chose the last 30 frames of each pure water simulation of each receptor as input for MaxCluster to determine representative structures. Before determining the heavy atom RMSD using an in-house python routine, we superimposed the obtained structures.

The simulations to determine the hydration sites were performed using the WATsite 3.0 protocol [41,42] that comes as a PyMol plugin [71]. The prepared structures of the eight receptors were used as input for the simulations and, since WATsite requires information about the location binding site, an AR ligand molecule known to bind both AF-2 and BF-3 was derived from a crystal structure (PDB ID: 2YLP) and superimposed to be located in the allosteric sites of the respective receptor. For each binding site, a separate simulation with an equilibration phase of 2 ns and a production stage of 20 ns at 298.15 K was run, totaling 352 ns of simulation time. We took the default timestep of 2.0 fs and frames with atomic coordinates collected every 2.0 ps. Long-range interactions were treated with the Particle Mesh Ewald method, non-bonded interactions were cut off at 10 Å, and heavy atoms were restrained with a spring constant of 2.5 kcal/mol/Å${}^{2}$. In the post-processing stage, we selected the DBSCAN clustering algorithm to determine the hydration sites and their occupancy. The backbone RMSD of the simulations was assessed using an in-house python routine.

### 3.5. Crystal Structure Analysis

For each receptor, all crystal structures with a resolution below 2.5 Å were retrieved from the Protein Data Bank and superimposed. Next, only the water molecules were kept in the structures and merged into a single PDB file that was used as input for an in-house python routine that determined the cluster centroids along with their occupancy using the DBSCAN algorithm with an epsilon value of 0.9 and *n* set to 2.0, similar to other protocols [72]. Clusters fulfilling the selected minimal occupancy criterion, depending on the number of input structures (Appendix A), were considered as conserved and further compared to the prediction from WATsite using a distance threshold of 1.4 Å to establish a consensus between the two approaches.

### 3.6. Molecular Docking

We used the Glide protocol [57,58] to dock the prepared ligands into the AF-2 and BF-3 sites of the selected panel of NRs. In the Recepor Grid Generation panel within Maestro, we defined the cubic grid box to be located at either site, with an inner box size of 10 Å and an outer box size of 22.4 Å. In order to define the binding site, we superimposed an AR ligand molecule on each receptor in the Protein Structure Alignment panel [63]. All actives were grouped according to their target site and receptor before they were docked, using both SP and XP protocols of Glide, to all sites in the set. In addition, we redocked known cocrystallized ligands and calculated the RMSD to the native pose in the Superposition panel in Maestro. In order to assess the reliability of the SP docking protocol, we further generated a decoy dataset for each compound group using the DUD-E webserver based on SMILES codes [73]. The ROC AUC metric, which characterizes if a randomly chosen known active molecule will rank higher than a randomly chosen decoy, was measured in the Enrichment Calculator within Maestro as described in detail in our previous work [63,74]. Crystal mates were visually inspected in Maestro.

### 3.7. Data Availability

All described in-house python routines and files are available in our public repository at https://github.com/mmodbasel/NR_allosteric_sites. Structure files with the determined hydration sites using both WATsite and the crystal structure analysis are available in PDB format together with files that allow the cosolvent densities to be viewed in CNS format, which is compatible with PyMol.

## 4. Conclusions

Several allosteric inhibitors for the AR proved their efficacy in blocking AR signaling through experiments with cells or xenograft in vivo studies [2,8,17]. Besides limitations in potency, the clinical application of this interesting compound class is hampered by off-target binding due to high sequence identity among hormonal NRs. From this viewpoint, the BF-3 site displayed advantages over the AF-2 site as a drug target in our analysis focused on sequence identity, pharmacophores, and hydration sites. Further, we recommend intensive selectivity testing to a wider array of NRs, particularly when inhibitors are targeting the AF-2 site, based on our results. Differences in probe densities reported in this study might be exploited to rationally design novel compounds and give insight into important structure–activity relationships. In our Appendix A, we provide the complete density maps obtained from the cosolvent simulations that can, for example, be incorporated in a pharmacophore-based screening campaign. Importantly, certain therapeutic scenarios might benefit by the concurrent binding to multiple NRs as we discussed regarding ERα inhibitors. In addition, future studies will need to consider potential synergistic effects of the simultaneous administration of orthosteric and allosteric inhibitors as well as combinations of inhibitors targeting the AF-2 and BF-3 sites concurrently. In our hydration site analysis, we identified water molecules that were conserved among multiple receptors including a reoccurring network of water molecules that formed an enthalpically favorable first-shell hydration layer around inhibitors at the BF-3 site. By means of the provided Appendix A, the gain in desolvation free energy for a particular ligand can be estimated by accounting for the displaced waters in the bound state. By docking a large set of allosteric inhibitors, we demonstrated a modest accuracy of the applied protocol and suggest the inclusion of water molecules and protein flexibility into future predictions. Additionally, we suggest residues that could be considered in flexible docking calculations based on a quantification of the per-residue conformational adaptation in the presence of different cosolvent molecules. In conclusion, this work provides a foundation to refine both selectivity and potency of allosteric inhibitors in a rational manner. Improving these properties will likely increase the therapeutic applicability of this interesting compound class.

## Figures and Tables

**Figure 1 ijms-21-00534-f001:**
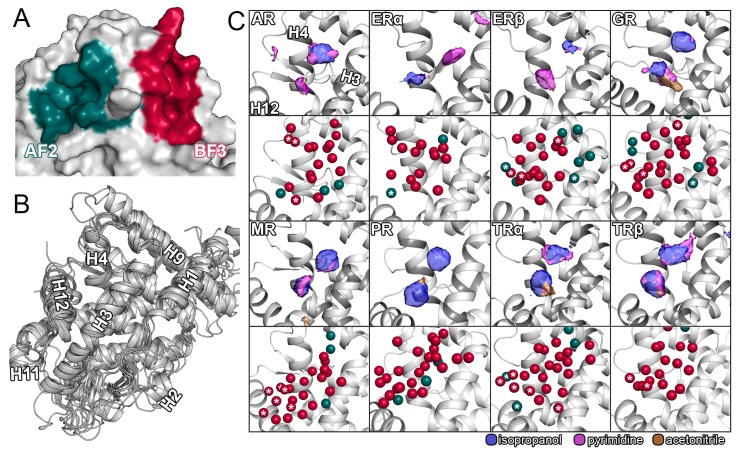
Structural overview, results from cosolvent simulations, and hydration site prediction for the AF-2 site. (**A**) AF-2 and BF-3 sites of the androgen receptor (PDB ID: 3L3X). (**B**) Structural alignment of AR, ERα, ERβ, GR, MR, PR, TRα, and TRα. Secondary structure elements were assigned according to Tan and colleagues [44]. (**C**) For each receptor, the results of cosolvent simulations (upper part) and hydration site prediction (lower part) from WATSite for the AF-2 site are given. The color scheme for the cosolvent densities is given below the figure. The densities are shown at an isovalue of 12. Water molecules, that were found to be conserved based on the crystal structure analysis were colored in pine green and water molecules with a negative enthalpy (ΔH < −1.0 kcal/mol) were indicated with asterisks.

**Figure 2 ijms-21-00534-f002:**
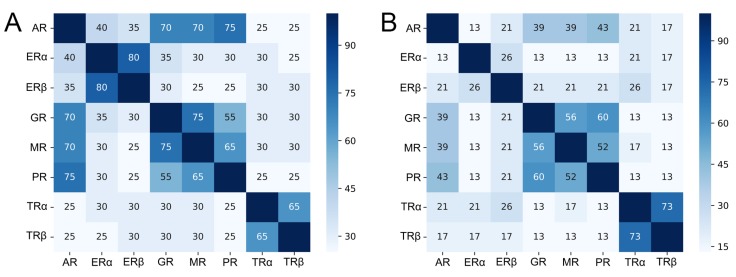
Sequence identity analysis of residues in (**A**) the AF-2 and (**B**) the BF-3 sites. The identity is given as a percentage of the maximally achievable score based on the considered residues.

**Figure 3 ijms-21-00534-f003:**
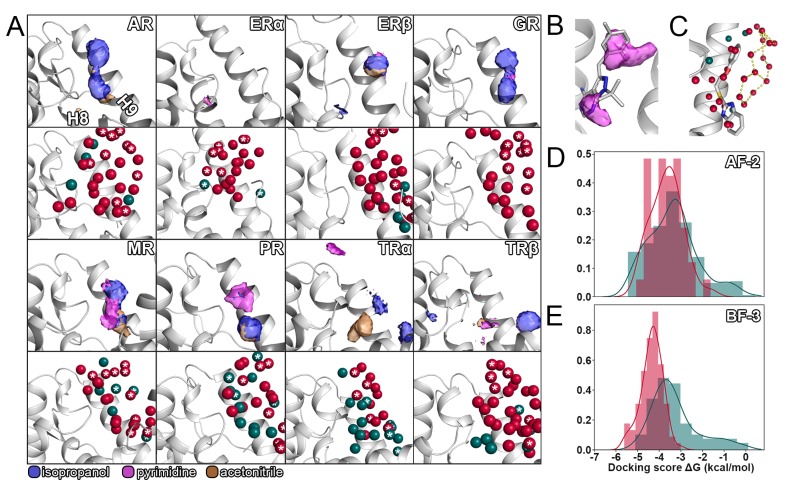
Results from cosolvent simulations, hydration site prediction for the BF-3 site, and molecular docking. (**A**) For each receptor, the results of cosolvent simulations (upper part) and hydration site prediction (lower part) from WATsite for the AF-2 site are given. The color scheme for the cosolvent densities is given below the figure. The densities are shown at an isovalue of 12. Water molecules, that were found to be conserved based on a crystal structure analysis were colored in pine green and water molecules with a negative enthalpy (ΔH < −1.0 kcal/mol) were indicated with asterisks. (**B**) Density of pyrimidine at the AF-2 overlapping with cocrystallized ligand (PDB ID: 2PIP). (**C**) Cluster of water molecules at the BF-3 of the androgen receptor (AR). A cocrystallized ligand molecule is shown for comparison (PDB ID: 4HLW). Polar contacts were visualized in PyMol. (**D**) Distribution of docking scores of AR AF-2 inhibitors. Confirmed actives are shown in red, while the remaining compounds of the library are colored pine green. (**E**) Distribution of docking scores of AR BF-3 inhibitors.

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
