# Peer review of "Allosteric Binding Sites On Nuclear Receptors: Focus On Drug Efficacy and Selectivity"

_ijms, 2020, doi:10.3390/ijms21020534_

Round 1
Reviewer 1 Report
The work by Fischer and Smieško focuses on allostery of nuclear receptors and assesses “druggability” of two different allosteric bonding sites on nuclear receptors using computational approaches: atomistic molecular dynamics (MD) simulations, including the cosolvent approaches, and molecular docking. The manuscript is well designed and written, and the results are likely to be important for oncology drug discovery community. There are some issues, however, that need to be addressed before the manuscript can be accepted:
The “resistance mechanisms”, which are mentioned in the introduction (p.1, l.23), may include alternative splicing and cleaving off the entire ligand-binding domain of the receptor (check, for instance, Duan et al., Nucleic Acids Res. 2019 Dec 16;47(22):11623-11636). This is the case of androgen receptor and it is the major cause of treatment resistance. While this does not invalidate the study, which focuses on the ligand-binding domain, it should be mentioned – either in the introduction, or in the discussion.My serious concern is the duration of trajectories: the simulations should be extended. 40 ns of pure water simulations, 20 ns of trajectories to determine the site hydration, and 5 ns cosolvent simulations should be increased at least 3-4 times to draw meaningful conclusions on conformational changes coupled to allosteric mechanisms.
I am also concerned by poor reproducibility of native binding poses (supplementary figures S9-S12), commented in the results and discussion (2.5). I would recommend authors to try different scoring functions (different docking software, perhaps, as well?), and the follow-up by the MD simulation (~50 ns), to evaluate the stability of the binding mode predicted by molecular docking. The latter should be done anyway, for the best scoring allosteric ligands.
Author Response
Comment 1: The “resistance mechanisms”, which are mentioned in the introduction (p.1, l.23), may include alternative splicing and cleaving off the entire ligand-binding domain of the receptor (check, for instance, Duan et al., Nucleic Acids Res. 2019 Dec 16;47(22):11623-11636). This is the case of androgen receptor and it is the major cause of treatment resistance. While this does not invalidate the study, which focuses on the ligand-binding domain, it should be mentioned – either in the introduction, or in the discussion.
We are aware of constitutively active androgen rececptor (AR) splice variants lacking the ligand binding domain (LBD). We initially did not discuss these resistance mechanisms since they mainly affect one of the eight receptors assessed in our study. However, we agree that they should be mentioned to give a complete picture since patients suffering from the mentioned resistance mechanisms would likely not benefit the same way from NR allosteric inhibitors targeting the LBD.
We therefore added the following statement to the beginning of the Results/Discussion section in order to address this issue:
“Even though constitutively active splice variants of the AR lacking the LBD regularly arise in late stages of the disease, the AR LBD remains a drug target of high interest, especially in early stages of pharmacological treatment.”
Comment 2: My serious concern is the duration of trajectories: the simulations should be extended. 40 ns of pure water simulations, 20 ns of trajectories to determine the site hydration, and 5 ns cosolvent simulations should be increased at least 3-4 times to draw meaningful conclusions on conformational changes coupled to allosteric mechanisms.
Please note that our study does not have the ambition to address the allosteric mechanism on how the AF-2 site or the BF-3 site regulate NR signaling, but is rather focused on the drug design strategy and selectivity aspect of these sites. The molecular mechanism of allosteric regulation has already been described and discussed in previous studies [1, 10].
Besides movements of side-chains on the protein surface in response to different cosolvent molecules, known to happen on the picosecond time scale [2], we did not discuss any conformational changes in our manuscript. However, we agree that differences between the apo and holo cosolvent MD simulations might have evolved in longer simulations as we mentioned in the Results/Discussion section of the original manuscript:
“[…] we did not observe significant differences regarding the cosolvent densities between our apo and holo simulations (Figure S3 and S4). Potentially, a protocol with prolonged individual simulations or the application of biasing potentials might induce more pronounced changes, since conformational adaptations affecting the surface of the receptor have to occur over a long distance and naturally require substantial simulation efforts.”
Along with the reasons we state below, we therefore trust that our simulation protocols are appropriate for the individual readouts and conclusions.
We performed cosolvent MD simulations according to a recommended established workflow [3,4,5] as it is included in the Desmond simulation suite. In this protocol, each cosolvent system is simulated for 20 ns (15 ns equilibration followed by 5 ns sampling phase), which results in a cumulative simulation time of 600 ns per receptor, since 10 replica simulations are performed and evaluated for each of the three selected probe molecules. The purpose of these simulations was to determine density-based pharmacophores rather than assessing conformational changes coupled to the allosteric regulation of the receptor or the induction of cryptic subpockets. Previous studies benchmarked the duration of cosolvent MD simulations and came to the conclusion that short simulations (as we performed them) are sufficient to induce conformational change such as the induction of cryptic binding pockets [5], which even goes beyond the scope of our study. Further, it was suggested that longer simulations may result in the denaturation of the protein or phase separation of the probe molecules [6,7]. We validated the simulations in terms of RMSD analyses and show the data in Tables S1-S8.
The simulations in “pure water” were performed as a baseline, to assess effects of probe molecules on the protein structure. Therefore, for consistency, these simulations were performed in a comparable time scale (40 ns) as the cosolvent MD simulations (20 ns). Still, we slightly increased the simulation time compared to the cosolvent simulations in order to ensure proper sampling and convergence. Regarding convergence, we added the RMSD of these simulations to the supporting information (Figure S7).
Regarding the simulations to determine the hydration sites (WATsite), we used the standard recommended simulation duration of 20 ns, since the developers of the routine proved 10 ns MD simulations to be sufficient for convergence of the hydration sites in a recent study [8] – even for occluded binding pockets. This is, to some extent, associated with the restraints that are applied on the protein atoms (as shown in the RMSD analysis in the newly added Figure S10).
Last but not least, prolonging all simulations (currently over 10 microseconds) by at least 3-4 times would result in a massive computational cost (30-40 microseconds), definitely not feasible in the given time frame of this peer review and exceeding our current GPU-accelerated simulation capabilities. The assessment of such distinct conformational changes caused by the presence of the ligand in the binding pocket will be investigated in a future dedicated study focused on a few selected target(s).
Comment 3: I am also concerned by poor reproducibility of native binding poses (supplementary figures S9-S12), commented in the results and discussion (2.5). I would recommend authors to try different scoring functions (different docking software, perhaps, as well?), and the follow-up by the MD simulation (~50 ns), to evaluate the stability of the binding mode predicted by molecular docking. The latter should be done anyway, for the best scoring allosteric ligands.
We agree that the results of redocking (“reproducibility”) may not be the most accurate, as we discussed in the original version of the article where we also provide possible explanations (e.g. crystal mates) for the observed discrepancies. As you requested, we performed an additional docking run using an alternative scoring function (Glide XP) and included the results (regarding score distribution) in the supporting information (Figure S17). These additional results further support our original conclusions as we now also state in the revised manuscript.
However, your request to perform MD simulations to post-process the obtained docking poses would only be applicable to the redocking procedure and would therefore not add any value to our cross-docking analysis, which is based on docking of the whole ligand library and the resulting score distribution. Performing 50 ns MD simulations for all our poses (334 compounds * 8 receptors * 2 sites) would again require enormous simulation time (267.2 microseconds) and would be by far out of our computational capabilities.
Our evaluation based on the area under the curve of the receiver operation characteristic (ROC AUC) proved the algorithm to be sensitive for the study purpose, which was essentially to distinguish binders from non-binders.
In addition, the docking protocol as we used it (Schrodinger Glide SP) was previously applied in multiple studies that led to successful discovery of allosteric NR inhibitors [9-12].
Reviewer 2 Report
The manuscript describes an in silico approach for investigating the AF-2 and BF-3 sites of nuclear receptors with respect to ligand binding using molecular dynamics simulations and molecular docking approaches.
It is very difficult to extract the relevant information from the study as it is hidden in a lot of text containing general information. Basically, this is the major drawback of this study. In my opinion, the analysis is too superficial and not going in-depth which is also evident by just three figures in the manuscript. The docking part does not very well fit to the simulation work and the conclusions drawn from it are pretty vague. Also the docking results should have been investigated using MD simulations.
In my opinion, the manuscript, after a thorough revision, much better fits to a journal focusing on computational studies such as Journal of Molecular Modeling or similar.
Minor comments
It is unclear how each simulation was evaluated for validity, e.g. backbone RMSD, energies.
I don’t see how the content of Figure S2&3 can be denoted as pharmacophore as written in the header, also the legend of the colors is missing (the caption states its in the main text, but I couldn’t find it there).
Figure 1B: different colors should help to distinguish the different superposed structures.
Figure S1 & S2: the surface representation on the right does not help much here. One cannot really see a pocket. The representation should be significantly improved (e.g. less shiny and coloring as the depiction on the left) or removed
Author Response
Reviewer 2
Comment 1: It is very difficult to extract the relevant information from the study as it is hidden in a lot of text containing general information. Basically, this is the major drawback of this study. In my opinion, the analysis is too superficial and not going in-depth which is also evident by just three figures in the manuscript.
In our original manuscript, we discuss the most stunning differences within our selection of hormonal nuclear receptors. Discussing everything in fine detail is not feasible if we want to keep the format of a regular research article – Therefore we rather provide detailed supplementary materials including an online repository (GitHub) featuring the cosolvent densities, as well as all 3D positions of determined water molecules. This is also referred to in the conclusion of the main text. For example, the data about the water molecules allows to score ligands with an additional desolvation term and offers potential to improve the design of novel ligands.
While preparing the manuscript we carefully considered which information needs to be presented in a pictorial form as we did not want to overflow readers with too much content in the main article. Our Figures are logically ordered and divided into focused sub-figures to optimally use the given space. Our supporting materials include a high number of figures as well as tabular data for readers desiring to go “in-depth”.
Comment 2: The docking part does not very well fit to the simulation work and the conclusions drawn from it are pretty vague. Also the docking results should have been investigated using MD simulations.
The conclusions we make on based on the docking score distribution are supported by docking an appropriate library of decoys, as it became common practice. Essentially, we confirm the BF-3 site to be a more selective drug target based on the distribution of the docking scores. Since interactions with the “charge clamps” of the allosteric sites were previously suggested to be involved as a selectivity factor, we investigated several binding poses and detected a potentially important interaction correlating with reported experimental affinity measurements. The remaining data we discuss in this paragraph (first part) refers to the validation of the docking protocol. As it was requested by Reviewer 1, we performed additional docking runs using the Glide XP protocol and could further confirm our conclusions regarding selectivity (cf. Figure S17).
Your request to perform MD simulations to post-process the obtained docking poses, however, would only be applicable to the redocking procedure and therefore would not add much value to our cross-docking analysis, which is based on docking of the whole ligand library and the resulting score distribution. Performing MD simulations to post-process all poses (334 compounds * 8 receptors * 2 sites), typically chosen to run for 50 ns, would require enormous simulation time (267.2 microseconds) and would be by far out of our computational capabilities.
Comment 3: In my opinion, the manuscript, after a thorough revision, much better fits to a journal focusing on computational studies such as Journal of Molecular Modeling or similar.
This manuscript was written upon invitation from the IJMS for the special issue “Nuclear Receptors 2.0” and the contents of our manuscript were pre-approved by the editorial office of the journal. Since the scope of the journal includes computational studies, we trust that our manuscript will attract interest of the large readership of the IJMS. Furthermore, our manuscript represents not a purely computational work, but was rather designed with tight link to the existing experimental data which are properly referenced.
Comment 4: It is unclear how each simulation was evaluated for validity, e.g. backbone RMSD, energies.
Due to your concern, we performed an RMSD-based analysis for all simulations (pure water, cosolvent, and WATsite) and added the respective data to the supporting information (Figures S7, S10, and Tables S1-S8). We now discuss the evaluation of the simulations in the respective paragraph in the main text. The RMSD analysis of the cosolvent MD simulations (based on the last frame provided by the Desmond routine) rarely presented values above 2.0 Å indicating minor “deviations” in the protein structure. The WATsite simulations are based on a protocol with restrained protein atoms resulting in a small RMSD of around 0.20 Å for every simulation. Regarding the simulations in pure water, the backbone RMSD analysis (Figure S7) of all triplicates simulations in pure water revealed excellent (AR, ERα, GR, PR, TRβ) to sufficient (ERβ, MR, TRα) convergence. Since the only readout we make based on these simulations is the adaptation of side chains based on appropriately selected representative structures, we now comment this issue in the respective paragraph of the revised manuscript.
Comment 5: I don’t see how the content of Figure S2&3 can be denoted as pharmacophore as written in the header, also the legend of the colors is missing (the caption states its in the main text, but I couldn’t find it there).
Please not that the header you are referring to is meant as the heading of the chapter – we kept the chapter heading consistent to the ones used in the article and this figure appears to be referenced in the chapter “Distinct Pharmacophores of the Allosteric Sites”. We agree that the way we use the term pharmacophore might confuse readers referring to its classical use (as it is e.g. used in pharmacophore-based screening methods). We therefore tried to clarify this by stating in the respective paragraph:
“The density maps of the BF-3 site of both TRs substantially differ from the ones of other receptors, which reduces the odds for the cross-binding of compounds harboring the proposed density-based pharmacophores.“
Thank you for noticing the shortcoming regarding the legend of these figures. We did not intend to refer to the main “text”, but rather the figures showing probe densities in the main article, where we provided the respective legend. To avoid this confusion and hopefully improve readability, we added the same legend to Figure S2 and S3.
Comment 6: Figure 1B: different colors should help to distinguish the different superposed structures.
We thought the same and created the figure with multiple colors. Unfortunately, the figure gets very busy and unreadable in this case (c.f. below). In our opinion, the current figure still allows a structural overview of the assessed receptors and we added it to highlight the similar fold among this set of NRs. Based on such a single alignment figure, we do not expect the reader to compare structures among the set of receptors.
Comment 7: Figure S1 & S2: the surface representation on the right does not help much here. One cannot really see a pocket. The representation should be significantly improved (e.g. less shiny and coloring as the depiction on the left) or removed.
We agree the mentioned representation was not optimal. We therefore changed the surface coloring strategy to only two colors showing the location of the sites and allow the reader to compare them regarding surface topology – by again choosing the same viewing angle as on the “left side”, this completes the structural overview of the sites.

Reviewer 3 Report
This is a very interesting paper in which the authors investigate the role of allostery in the binding of drugs acting at the level of 8 hormonal nuclear receptors. This is a hot topic in biology, considering that inactivation of nuclear receptors result in the cure of oncologic, metabolic, reproductive and immunologic diseases. The paper is very well written and technically well done. It also reports comparison with available experimental data, sustaining their overarching hypothesis of a critical role of the signal transduction. Hence, I recommend publication upon minor revisions.
The importance of allostery in biochemistry and, more specifically, in protein/nucleic acid complexes has been widely proven. However, the authors do not discuss how this phenomenon has proven to be the driving underlying force for the drug inhibition and regulation of protein nucleic acid complexes. The authors should provide a comprehensive explanation of how allostery is at the basis of drug-drug synergy and of the regulation in protein/nucleic acid complexes, such as the nuclear receptors (see: Adhireksan et al. Nat. Commun. 2017, 8, 14860; Palermo et al. JACS 2017, 2017, 139, 16028). How do these well-established allosteric routes compare with the mechanism proposed by the authors? This is a key point that should be discussed prior publication.
An important missing point concerns the analysis methods used by the authors. Over the years, accurate network models and simulation methods have shown to provide clear explanation of the allosteric route, enabling also to harness allostery for drug discovery. The authors should discuss their methodology employed and provide more information on state-of-the-art analysis tools employed to investigate allostery (missing reference: Wodak et al. Structure 2019, 4, 566-578). This is an important point to enable reproducibility and to place the article in a broader scenario.
Author Response
Reviewer 3
Comment 1: The importance of allostery in biochemistry and, more specifically, in protein/nucleic acid complexes has been widely proven. However, the authors do not discuss how this phenomenon has proven to be the driving underlying force for the drug inhibition and regulation of protein nucleic acid complexes. The authors should provide a comprehensive explanation of how allostery is at the basis of drug-drug synergy and of the regulation in protein/nucleic acid complexes, such as the nuclear receptors (see: Adhireksan et al. Nat. Commun. 2017, 8, 14860; Palermo et al. JACS 2017, 2017, 139, 16028). How do these well-established allosteric routes compare with the mechanism proposed by the authors? This is a key point that should be discussed prior publication.
Thank you for your thought-provoking comment. We agree to have missed touching potential synergistic affects of concurrent binding to both sites in our study. In the case of nuclear receptors, it is known that binding to the orthosteric site is coupled distinct conformational changes on the surface of the receptor influencing the conformation of the AF-2 site. We mention this in the following statement (2.3. Conformational Changes of the Allosteric Sites):
“Even though it was suggested that the association of allosteric inhibitors is dependent on the presence of an agonist in the orthosteric site […] For example, association of inhibitors to the LBP has been shown to structurally modulate the AF-2 and its capability to interact with coactivator proteins, mainly by conformational change of helix-12 (H12).”
We agree that this is relatively “thin” for such an important topic and that we missed to discuss potential synergy among inhibitors of the allosteric sites themselves. Since combination therapy is regularly applied in the pharmacological treatment of various forms of cancer, we now discuss potential synergistic effects in the respective paragraph:
“Combination therapy with multiple drugs is regularly applied in cancer pharmacotherapy and therefore potential synergistic effects of allosteric and orthosteric inhibitors will have to be considered in future studies. Likewise, the simultaneous treatment with AF-2 and BF-3 inhibitors might produce mixed results, since binding of inhibitors to the BF-3 site is known to reduce the affinity of coactivator peptides in an allosteric mechanism and might affect a potential drug-drug synergy.”
Also, we added more context on this into the Conclusion, with an outlook for potential future studies:
“In addition, future studies will have to consider potential synergistic effects of the simultaneous administration of orthosteric and allosteric inhibitors as well as combinations of inhibitors targeting the AF-2 and BF-3 sites concurrently.”
The mechanistic investigation of a potential drug synergy would go beyond the scope of this study which focused on the selectivity and drug design strategy of the allosteric inhibitors compared among the selected set of hormonal nuclear receptors. We are currently working on a follow-up study specifically focused on drug combinations requiring a large number of MD simulations in the microsecond timescale, which would not be feasible in the context of this peer review process (limited to a few days).
On the major note, we did not investigate the allosteric regulation of either site in our manuscript and therefore we did not propose any mechanism. The mechanism of nuclear receptor allosteric inhibitors was already shown and discussed in various publications. Inhibitors of the AF-2 site prevent the binding of coactivator peptides necessary for downstream signaling (dimerization, nuclear translocation, DNA-binding, …). On the other hand, binding to the BF-3 site has been shown to modulate the binding of coactivator peptides to the AF-2 site in an allosteric mechanism. In this regard, individual residues involved (domino mechanism) in the allosteric regulation of the AF-2 site through BF-3 inhibitors were already suggested [1]. We generally report on this in the introduction section of the original manuscript:
“The AF-2 site corresponds to a protein-protein interaction surface for the binding of coactivator proteins essential for downstream signaling which renders it an attractive target for potential inhibitors. While the BF-3 site has been initially shown to allosterically regulate binding of coactivators to the AF-2 site [2,6–8], it has been suggested as interaction surface for the engagement with chaperones that associate NRs [2,9,10].”
Comment 2: An important missing point concerns the analysis methods used by the authors. Over the years, accurate network models and simulation methods have shown to provide clear explanation of the allosteric route, enabling also to harness allostery for drug discovery. The authors should discuss their methodology employed and provide more information on state-of-the-art analysis tools employed to investigate allostery (missing reference: Wodak et al. Structure 2019, 4, 566-578). This is an important point to enable reproducibility and to place the article in a broader scenario.
As we mention in our answer to your first comment, our work is not focused on the allosteric mechanism per se, since this was already presented and discussed in various previous publications [1, 10, 11]. Our work is rather focused on drug design aspects and selectivity. We strongly trust that the reproducibility of our results is properly given based on a detailed “Materials and Methods” section.
Answer Letter References
[1] Estebanez-Perpina, E.; Arnold, L. A.; Nguyen, P.; Rodrigues, E. D.; Mar, E.; Bateman, R.; Pallai, P.; Shokat, K. M.; Baxter, J. D.; Guy, R. K.; Webb, P.; Fletterick, R. J. A Surface on the Androgen Receptor That Allosterically Regulates Coactivator Binding. Proc. Natl. Acad. Sci. 2007, 104 (41), 16074–16079.
[2] Zwier, M. C.; Chong, L. T. Reaching Biological Timescales with All-Atom Molecular Dynamics Simulations. Curr. Opin. Pharmacol. 2010, 10 (6), 745–752.
[3] Ghanakota, P.; Van Vlijmen, H.; Sherman, W.; Beuming, T. Large-Scale Validation of Mixed-Solvent Simulations to Assess Hotspots at Protein-Protein Interaction Interfaces. J. Chem. Inf. Model. 2018, 58 (4), 784–793.
[4] Ghanakota, P.; Carlson, H. A. Moving beyond Active-Site Detection: MixMD Applied to Allosteric Systems. J. Phys. Chem. B 2016, 120 (33), 8685–8695.
[5] Tan, Y. S.; Spring, D. R.; Abell, C.; Verma, C. S. The Application of Ligand-Mapping Molecular Dynamics Simulations to the Rational Design of Peptidic Modulators of Protein-Protein Interactions. J. Chem. Theory Comput. 2015, 11 (7), 3199–3210.
[6] Zariquiey, F. S.; Souza, J. V. De; Bronowska, A. K. Cosolvent Analysis Toolkit ( CAT ): A Robust Hotspot Identification Platform for Cosolvent Simulations of Proteins to Expand the Druggable Proteome. 2019, 1–14.
[7] Lexa, K. W.; Carlson, H. A. Full Protein Flexibility Is Essential for Proper Hot-Spot Mapping. J. Am. Chem. Soc. 2011, 133 (2), 200–202.
[8] Masters, M. R.; Mahmoud, A. H.; Yang, Y.; Lill, M. A. Efficient and Accurate Hydration Site Profiling for Enclosed Binding Sites. J. Chem. Inf. Model. 2018, 58 (11), 2183–2188.
[9] Axerio-Cilies, P.; Lack, N. A.; Nayana, M. R. S.; Chan, K. H.; Yeung, A.; Leblanc, E.; Guns, E. S. T.; Rennie, P. S.; Cherkasov, A. Inhibitors of Androgen Receptor Activation Function-2 (AF2) Site Identified through Virtual Screening. J. Med. Chem. 2011, 54 (18), 6197–6205.
[10] Jehle, K.; Cato, L.; Neeb, A.; Muhle-Goll, C.; Jung, N.; Smith, E. W.; Buzon, V.; Carbó, L. R.; Esteb́anez-Perpiñ́a, E.; Schmitz, K.; Fruk, L.; Luy, B.; Chen, Y.; Cox, M. B.; Bras̈e, S.; Brown, M.; Cato, A. C. B. Coregulator Control of Androgen Receptor Action by a Novel Nuclear Receptor-Binding Motif. J. Biol. Chem. 2014, 289 (13), 8839–8851.
[11] Martinez-Ariza, G.; Hulme, C. Recent Advances in Allosteric Androgen Receptor Inhibitors for the Potential Treatment of Castration-Resistant Prostate Cancer. Pharm. Pat. Anal. 2015, 4 (5), 387–402.

Round 2
Reviewer 2 Report
In the present form the revised manuscript can be accepted